

# An online intelligent detection method for slurry density in concept drift data streams based on collaborative computing

Lanhao Wang[1], Hao Wang[2], Taojie Wei[3], Wei Dai[2,3] and Hongyan Wang[2]

[1] National Engineering Research Center of Coal Preparation and Purification, China University of Mining Technology, XuZhou, Jiangsu, China
[2] Artificial Intelligence Research Institute, China University of Mining and Technology, XuZhou, Jiangsu, China
[3] School of Information and Control Engineering, China University of Mining and Technology, XuZhou, Jiangsu, China

## ABSTRACT

In industrial environments, slurry density detection models often suffer from performance degradation due to concept drift. To address this, this article proposes an intelligent detection method tailored for slurry density in concept drift data streams. The method begins by building a model using Gaussian process regression (GPR) combined with regularized stochastic configuration. A sliding window-based online GPR is then applied to update the linear model's parameters, while a forgetting mechanism enables online recursive updates for the nonlinear model. Network pruning and stochastic configuration techniques dynamically adjust the nonlinear model's structure. These approaches enhance the mechanistic model's ability to capture dynamic relationships and reduce the data-driven model's reliance on outdated data. By focusing on recent data to reflect current operating conditions, the method effectively mitigates concept drift in complex process data. Additionally, the method is applied in industrial settings through collaborative computing, ensuring real-time slurry density detection and model adaptability. Experimental results on industrial data show that the proposed method outperforms other algorithms in all density estimation metrics, significantly improving slurry density detection accuracy.

# INTRODUCTION

The mineral processing workflow comprises several stages, including raw ore transportation, crushing and screening, grinding and classification, beneficiation, and dewatering (*Hodouin et al., 2001*). Among these, grinding and classification serve as a critical link between crushing and beneficiation, significantly influencing the overall workflow. Key equipment in grinding operations includes ball mills and hydrocyclones, whose performance directly affects grinding efficiency (*Mukhitdinov et al., 2024*; *Bradley, 2013*). The hydrocyclone feed density is a vital parameter impacting its overflow particle

Corresponding author
Hongyan Wang,
tbh344@cumt.edu.cn

size. Higher feed density increases slurry viscosity and resistance, resulting in coarser overflow particles and reduced classification efficiency. On the other hand, lower feed density improves classification efficiency but reduces throughput while increasing water and power consumption. Therefore, accurate monitoring and control of hydrocyclone feed density are essential for optimizing grinding and classification efficiency (*Reddy et al., 2023*).

Slurry density is a key metric in grinding and classification, directly influencing metal recovery rates, concentrate grades, production efficiency, and process stability (*Whitworth et al., 2022*). Current detection methods primarily rely on manual laboratory techniques and densitometers, with limited exploration of artificial intelligence (AI) applications. The pycnometer method is the most common manual technique, where a pycnometer is filled with slurry, weighed, and its density calculated using a formula. Densitometers, on the other hand, use precise instruments to determine material density based on physical principles. Recently, advances in AI technologies have enabled some innovative approaches for slurry density detection. For example, the combination of Prompt Gamma Neutron Activation Analysis (PGNAA) technology and artificial neural networks (ANN) has been proposed for online detection (*Huang et al., 2024*). Similarly, an approach based on closed-loop input error and deep learning offers a novel method for real-time slurry concentration prediction (*Han et al., 2024*).

In mineral processing, operational fluctuations such as variations in feed rate and water addition often lead to concept drift, causing slurry density detection models to degrade in performance (*Bayram, Ahmed & Kassler, 2022*). To address this issue, researchers have developed methods to enhance model adaptability to changing data distributions. These include selecting training samples that represent recent data distributions (*Fan, 2004*), employing online learning algorithms to update model parameters continuously, dynamically adjusting model structures for new data features (*Yang & Fong, 2015*), and applying weighted updates to reduce the influence of outdated data (*Sen, 2014*; *Martínez-Rego et al., 2011*). These techniques ensure model accuracy and adaptability in dynamic environments. This study investigates a modeling approach that combines mechanistic and data-driven methods to address the challenges of concept drift and meet the demands for accurate, real-time slurry density detection in mineral processing (*Cui et al., 2024*). We propose an online intelligent detection method for slurry density in concept drift data streams, leveraging collaborative computing. This approach is not limited to slurry density detection and can be extended to monitor other industrial process variables, enhancing the accuracy of industrial parameter detection and improving production efficiency (*Wang et al., 2023*).

## PROCESS DESCRIPTION AND CHARACTERISTICS ANALYSIS

Grinding and classification are among the most critical stages in mineral processing (*Yuan et al., 2020*). These stages typically involve a closed grinding circuit comprising ball mills, hydrocyclones, and slurry pumps. The primary grinding circuit includes a ball mill and a spiral classifier, while the secondary circuit consists of a ball mill, hydrocyclone, and pump

sump. In the primary circuit, ore is mixed with water and ground in the ball mill, after which the slurry is classified by the spiral classifier. Coarse particles are returned to the ball mill for further grinding, while finer particles proceed to the secondary circuit. In the sump, additional water is added, and the slurry is pumped into the hydrocyclone. The hydrocyclone uses centrifugal force to separate the slurry, discharging coarse particles for further grinding and sending finer particles to subsequent beneficiation processes (*Wang & Chai, 2019*).

The mechanistic analysis of slurry flow in pipelines focuses on selecting auxiliary variables and building a comprehensive model for slurry density detection (*Ma, Wang & Peng, 2024*). Resistance losses are categorized based on boundary conditions. For smooth boundaries, frictional resistance arises from boundary-fluid interactions and fluid viscosity. Local resistance losses occur due to sudden boundary changes, such as pipe bends, valves, or cross-sectional variations, which can alter flow paths and velocities, potentially causing vortices. Since slurry density detection is performed in vertical pipelines, local resistance losses are negligible, and only frictional resistance losses are included in calculations (*Peet, Sagaut & Charron, 2009*).

In an ideal scenario without resistance losses, the pressure difference is given by:

$$\Delta p = \rho g \Delta H. \tag{1}$$

In Eq. (1), $\rho$ is the slurry density, $g$ is gravitational acceleration, and $\Delta H$ is the height difference of the liquid surface. During slurry flow, frictional resistance losses occur, which are described by:

$$H_f = \gamma \frac{L}{D} \frac{V^2}{2g}. \tag{2}$$

In Eq. (2), $\gamma$ is the frictional resistance coefficient, $L$ is the pipe length, $D$ is the pipe diameter, $V$ is the average flow velocity, and $g$ is gravitational acceleration. In actual industrial processes, the total pressure difference $\Delta p = \rho g \Delta H - H_f$ can be expressed as:

$$\rho = \frac{\Delta p + H_f}{g \Delta H}. \tag{3}$$

In industrial production, density measurement commonly relies on pressure differential signals from sensors placed at different heights. However, directly using these signals as inputs for detection models may reduce accuracy (*Li et al., 2020*). According to Bernoulli's principle, the total pressure in a fluid remains constant; as flow velocity increases, static pressure decreases. Slurry pressure meters, however, measure only static pressure.

As inlet velocity rises, the dynamic pressure difference between two points also increases. Traditional pressure sensors convert pressure into electrical signals by inducing deformation in a force-sensitive element, which changes resistance in a Wheatstone bridge and generates a potential difference output (*Xu et al., 2018*). While effective for measuring static pressure differences, this method cannot capture dynamic pressure changes, potentially reducing measurement accuracy if differential signals are used directly in

density models. Additionally, system and random errors in pressure measurements necessitate corrections to the high-pressure side absolute pressure $p_H$ and the low-pressure side absolute pressure $p_L$. The pressure difference $\Delta p(t) = p_H(t) - p_L(t)$ measured at time $t$ is adjusted as follows:

$$\Delta p(t) = A p_H(t) - B p_L(t) + C + l_1(p_H(t), p_L(t)). \tag{4}$$

In Eq. (4), $A$ and $B$ are correction coefficients for high pressure and low pressure, respectively; $C$ is the offset term, and $l_1(\cdot)$ represents unknown nonlinear errors in pressure measurement. The average flow velocity $V$ has a nonlinear relationship with the slurry pump current $i$ and frequency $f$:

$$V(t) = l_2(f(t), i(t)). \tag{5}$$

This relationship can be expressed as follows:

$$\rho(t) = \rho_0(t) + \Delta \rho(t). \tag{6}$$

In Eq. (6),

$$\rho_0(t) = k_1 p_H(t) + k_2 p_L(t) + k_3$$
$$\Delta \rho(t) = l(p_H(t), p_L(t), f(t), i(t))$$
$$k_1 = \frac{A}{g\Delta H}, k_2 = -\frac{B}{g\Delta H}, k_3 = \frac{C}{g\Delta H} (g\Delta H > 0)$$
$$l(\cdot) = \frac{l_1(p_H(t), p_L(t))}{g\Delta H} + \frac{\gamma L l_2^2(f(t), i(t))}{2g^2 D\Delta H}$$

where $l(\cdot)$ includes measurement errors and unknown nonlinear terms in the slurry flow process.

## MECHANISM AND DATA-DRIVEN ONLINE INTELLIGENT DETECTION METHOD FOR SLURRY

### Classification and handling methods of concept drift

In industrial environments, slurry density detection models face the challenge of concept drift, which refers to dynamic changes in data distribution or characteristics over time. Concept drift often arises from external factors such as variations in raw material properties, production processes, or equipment aging. To maintain prediction accuracy, detection models must adapt continuously to these evolving conditions.

Concept drift is generally categorized as follows:

1) Sudden drift: This involves rapid and significant changes in data features over a short time, often caused by abrupt shifts in raw material properties, equipment failures, or emergency adjustments. Such changes can lead to sudden prediction errors, requiring models to quickly adapt.

2) Gradual drift: Gradual drift occurs when data features evolve slowly over time, such as equipment aging or long-term fine-tuning of process parameters. Although these

changes may not immediately affect data distribution, model performance will degrade if left unaddressed. Dynamic update mechanisms are commonly used to adapt to these gradual changes.

3) Incremental drift: This refers to stable, cumulative changes in data distribution, such as progressive variations in slurry concentration across production batches. While each change is small, the cumulative effect can shift the data distribution, necessitating models capable of incremental learning.

4) Recurrent drift: Recurrent drift arises from cyclical factors like periodic equipment cleaning or routine production adjustments. Handling this type of drift requires models to recognize and leverage cyclical patterns to make appropriate adjustments.

In industrial slurry density detection, concept drift is common and often involves multiple drift types coexisting, placing high demands on model robustness and adaptability. In this study, the dataset primarily exhibits sudden and gradual drift. Sudden drift arises from abrupt changes in raw material properties, equipment failures, or emergency operational adjustments, leading to rapid shifts in data features. Gradual drift, in contrast, involves slow changes over time due to equipment aging or minor adjustments in process parameters. To address these drift types, the proposed detection model incorporates a sliding window mechanism and a forgetting mechanism to dynamically update model parameters. For sudden drift, the sliding window mechanism focuses on recent data, discarding outdated information to enable quick adaptation to abrupt changes. The window size is dynamically adjusted to promptly capture new feature distributions during drift events. Recursive formulas are also used to update key parameters online, ensuring the model responds without delays. For gradual drift, the forgetting mechanism reduces the weight of historical data over time, enhancing the model's sensitivity to current data. By dynamically adjusting the forgetting factor, the model ensures smooth updates for gradual changes while avoiding overreactions to short-term fluctuations. By combining these mechanisms, the proposed model effectively handles diverse types of concept drift in complex industrial environments, significantly improving the accuracy and stability of slurry density detection.

## Establishing and calibrating the comprehensive model for slurry density

In dynamic data environments, concept drift occurs when the statistical properties of data change over time, posing challenges for density detection models. This article proposes an intelligent detection algorithm for streaming data, combining a mechanistic model based on Gaussian process regression (GPR) and a data-driven model (*Wei et al., 2022*) based on a regularized stochastic configuration (RSC) Network for offline learning (*Zhang & Wang, 2021*). Initially, a subset of the data is selected to establish the initial model. Subsequently, the linear and nonlinear models are updated with streaming data, and the results of both models are combined to obtain the final slurry density detection value (*Zhang et al., 2024*). As new samples arrive, the linear model parameters are updated online using a recursive formula, yielding a linear model estimate $\hat{\rho}_0$ and its variance $\sigma^2$. The nonlinear model's

output weights are updated online using the teacher signal $\Delta\bar{\rho}(\Delta\bar{\rho} = \rho - \hat{\rho}_0)$ and the variance of the linear estimate $\sigma^2$ as labels, without altering the model structure. This provides the nonlinear model's density estimate $\Delta\hat{\rho}$. If the estimate falls outside the confidence interval $[\Delta\bar{\rho} - 3\sigma, \Delta\bar{\rho} + 3\sigma]$, the nonlinear model structure is dynamically adjusted to improve generalization performance. Otherwise, the overall model is updated.

## Mechanism-based model using online Gaussian process regression

The mechanistic model, representing the linear component, is based on the physical principles of slurry flow in pipelines (*Lui, Liu & Xie, 2022*). Using Gaussian process regression with a sliding window mechanism (OGPRSWM), the linear model updates its parameters in real-time. This approach reduces the influence of outdated data, improves parameter estimation, and ensures the model remains accurate and up-to-date (*Gu, Fei & Sun, 2020*).

Initially, GPR is employed to identify the linear component of slurry density (*Cao et al., 2023*). When input data $x'_a(k)$ at time $t = k$ is provided, the probability distribution of the mechanism model's output can be obtained as follows:

$$P\big(\hat{y}_a(k)|x'_a(k), X'_a(k-1), Y_a(k-1)\big)$$
$$= \mathcal{N}\big(\sigma_n^{-2}x'_a(k)A^{-1}X'_aT(k-1)Y_a(k-1), x'_a(k)A^{-1}x'_aT(k)\big). \tag{7}$$

In Eq. (7), $A = \Sigma^{-1} + \sigma_n^{-2}X'_aT(k-1)X'_a(k-1)$, $N_t$ represents the total number of training samples in the data pool. $X_a(k-1)$ and $Y_a(k-1)$ denote the input and output data used for training the linear model up to a given time, respectively. $x_a(k-i_1)$ and $y_a(k-i_1)$ represent the input and output data for training the linear model at a specific time; $\hat{\rho}_0(k) = \hat{y}_a(k) = \sigma_n^{-2}x'_a(k)A^{-1}X'_aT(k-1)Y_a(k-1)$ is the estimated result of the linear model, $\sigma^2(k) = x'_a(k)A^{-1}x'_aT(k)$ is the variance estimated by the Gaussian process regression.

Subsequently, an online Gaussian process regression with a sliding window mechanism is applied. During the initialization phase, dataset $\{X_b(k-1), Y_b(k-1)\}$ is used to construct the initial linear model, and the posterior distribution of parameters is estimated using $N_0$, the number of training samples in the data pool. $X_b(k+i_2)$ and $Y_b(k+i_2)$ denote the input and output data for training the linear model up to time $k + i_2$, while $x_b(k+i_2)$ and $y_b(k+i_2)$ represent the input and output data at time $k + i_2$:

$$P\big(\hat{\theta}(k-1)|Y_b(k-1), X'_b(k-1)\big)$$
$$= \mathcal{N}\left(\frac{1}{\sigma_n^2}A^{-1}(k-1)X'_bT(k-1)Y_b(k-1), A^{-1}(k-1)\right). \tag{8}$$

In Eq. (8),

$$X'_b(k-1) = [x'_b T(k-N_0), \ldots, x'_b T(k-2), x'_b T(k-1)]^T$$
$$x'_b(k+i_2) = [p_H(k+i_2), p_L(k+i_2), 1]$$
$$A(k-1) = \Sigma^{-1} + \sigma_n^{-2} X'_b T(k-1) X'_b(k-1).$$

After adding new samples and discarding older historical data, dataset $\{X'_b(k-1), Y_b(k-1)\}$ is updated to $\{X''_b(k), Y'_b(k)\}$, The newly added sample is $\{x'_b(k), y_b(k)\}$, and the discarded sample is denoted as $\{X'_{drop}, Y_{drop}\}$; Following the update of the data samples, $\sigma_n^{-2}$ is updated to $\sigma_n'^{-2}$. The key issue in updating the linear model is to update $A^{-1}(k-1)$ to $A^{-1}(k)$, where $A(k)$ is:

$$A(k) = \Sigma^{-1} + \sigma_n'^{-2} X''_b T(k) X''_b(k)$$
$$= \Sigma^{-1} + \sigma_n'^{-2} \sigma_n^2 \left(A(k-1) - \Sigma^{-1}\right) + \sigma_n'^{-2} \left(x'_b T(k) x'_b(k) - X'_{drop} T X'_{drop}\right). \tag{9}$$

The computational complexity of $A(k)$ primarily stems from matrix multiplication, with the original expression having a complexity of $O(N_w^2 \times 3)$. Let $N_w$ represent the sliding window size; the complexity of the recursive computation is $O\left(\left(N_{drop}^2 + 1\right) \times 3\right)$. Based on $N_w^2 >> N_{drop}^2 + 1$, the derived recursive formula is used for online updates of the linear model to reduce computational load and improve efficiency. By performing matrix inversion on $A(k)$, the updated parameters are obtained, and the estimated density value and variance for the new incoming data sample $x'_{on,lin}(k+1)$ are calculated as follows:

$$\hat{\rho}_b(k+1) = \sigma_n^{-2} x'_b(k+1) A^{-1}(k) X''_b T(k) Y'_b(k)$$
$$\sigma^2(k+1) = x'_b(k+1) A^{-1}(k) x'_b T(k+1). \tag{10}$$

## Data-driven model based on online regularized stochastic configuration networks

This article presents a novel learning algorithm, the Forgetting Mechanism Regularized Stochastic Configuration (FMRSC) algorithm, to address concept drift and enable online learning for data-driven models based on Regularized Stochastic Configuration (RSC) Networks (*Luo et al., 2022*). Unlike the Online Sequential Stochastic Configuration (OSSC) algorithm (*Chen & Li, 2022*), the proposed FMRSC method processes streaming data without requiring the retraining of the entire historical dataset. It achieves this by integrating regularization and forgetting mechanisms into the OSSC algorithm. Additionally, FMRSC dynamically adjusts the model structure using network pruning and stochastic configuration to handle concept drift effectively. This approach leverages recent data, minimizes reliance on outdated information, and enhances processing efficiency and adaptability (*Dai, Liu & Wang, 2024*).

### Online parameter update strategy

The RSC algorithm, used as the data-driven method in this study, is an improved version of the Stochastic Configuration Network (SCN) (*Wang & Li, 2017*). By incorporating

regularization techniques, RSC effectively mitigates overfitting, producing more robust and generalized neural network models.

The output $h_L$ of the hidden layer node $L$ and the supervision mechanism $\xi_{L,q}$, $q = 1, 2$ are defined as follows:

$$h_L = \left[ g_L\left(\omega_L^T x_c(k-1) + b_L\right), g_L\left(\omega_L^T x_c(k-2) + b_L\right), \ldots, g_L\left(\omega_L^T x_c(k-N) + b_L\right) \right]^T \quad (11)$$

$$\xi_{L,q} = \frac{\left(e_{L-1,q}^T h_L\right)^2}{\gamma} - (1 - r - \mu_L)e_{L-1,q}^T e_{L-1,q}, q = 1, 2. \quad (12)$$

In Eqs. (11) and (12), $\gamma = (h_L^T \cdot h_L + 1/C)^2/(h_L^T \cdot h_L + 2/C)$. Given $1 - \varepsilon < r < 1$, let $\mu_L = (1-r)/(L+1)$ and $C$ be the regularization coefficients. Then, perform $T_{\max}$ stochastic configurations. In each configuration, randomly select the input weights $\omega_L$ and bias $t_i$ for the $L$th hidden layer node within a certain range, and compute $\xi_{L,q}, q = 1, 2$. If $\min \xi_{L,1}, \xi_{L,2} \geq 0$ is satisfied, store $\omega_L, t_i, \xi_{L,q}$; if none of the $\xi_{L,q}$ configurations meet the condition, choose a larger $r$ value and reconfigure. After completing the random configurations, select the $\omega_L$ and $b_L$ corresponding to the largest $\sum_{q=1}^{2} \xi_{L,q}$ as the input weights and bias for the $L$th node.

The estimated value of the data-driven model at time $t = k$ is expressed as:

$$\Delta \hat{\rho}(k) = \sum_{j=1}^{L} \beta_j g_j\left(\omega_j^T x_c(k) + b_j\right). \quad (13)$$

Next, the parameters are updated online using a forgetting mechanism. Given dataset $\{X_d(k + N_{str} - 1), Y_d(k + N_{str} - 1)\}$, during the initialization phase of the nonlinear model, $\{X_d(k-1), Y_d(k-1)\}$ is used to construct the initial nonlinear model. Here, $X_d(k + i_2)$ and $Y_d(k + i_2)$ denote the input and output data for training the nonlinear model up to time $k + i_2$, and $x_d(k + i_2)$ and $y_d(k + i_2)$ represent the input and output data at time $k + i_2$. If a regularized random configuration network with $L$ hidden layer nodes is constructed based on these $N_0$ sets of training data, the optimization objective for the output layer weights $\beta_{k-1}$ is as follows:

$$\beta_{k-1} = \arg \min_{\beta_{k-1}} \left( \|H_{k-1}\beta_{k-1} - Y_d(k-1)\|^2 + \frac{1}{C}\|\beta_{k-1}\|^2 \right). \quad (14)$$

Let $H_{k-1}$ represent the output of the hidden layer nodes of the nonlinear model initialized with training data from Group $N_0$. Let $C$ denote the regularization term coefficient. The solution can be obtained as follows:

$$\beta_{k-1} = \left( H_{k-1}{}^T H_{k-1} + \frac{E}{C} \right)^{-1} H_{k-1}{}^T Y_d(k-1). \quad (15)$$

When data from Group $(N_0 + 1)$th reaches the model, it is necessary to update the weights with the latest data. To mitigate the influence of past data on the model parameter updates, a forgetting mechanism has been introduced. Given the fixed structure of the

neural network and the constant weights of the input layer, the optimization objective for obtaining new output layer weights $\beta_k$ is as follows:

$$\beta_k = \arg\min_{\beta_k} \left( \theta_k \|H_{k-1}\beta_k - Y_d(k-1)\|^2 + \|h_k\beta_k - y_d(k)\|^2 + \frac{1}{C}\|\beta_k\|^2 \right). \tag{16}$$

Let $\theta_k$ represent the forgetting factor, $h_k$ represent the hidden layer node outputs calculated from the $(N_0 + 1)$th Group dataset, and $H_k = [H_{k-1}^T, h_k^T]^T$ be given.

The solution can be obtained as follows:

$$\beta_k = \left( H_k^T \Theta_k^T \Theta_k H_k + \frac{E}{C} \right)^{-1} H_k^T \Theta_k^T \Theta_k Y_d(k). \tag{17}$$

In Eq. (17), $\Theta_k = \text{diag}\{\sqrt{\theta_k}, \sqrt{\theta_k}, \ldots, \sqrt{\theta_k}, \sqrt{\theta_k}, 1\}$.

Let $P_{k-1} = H_{k-1}^T H_{k-1}$, $P_k = \theta_k^2 H_{k-1}^T H_{k-1} + h_k^T h_k = \theta_k^2 P_{k-1} + h_k^T h_k$, then we obtain:

$$\begin{aligned} \beta_k &= \left( P_k + \frac{E}{C} \right)^{-1} \left( \theta_k^2 \left( P_{k-1} + \frac{E}{C} \right) \beta_{k-1} + h_k^T y_d(k) \right) \\ &= \beta_{k-1} + \left( P_k + \frac{E}{C} \right)^{-1} \left( \frac{\theta_k^2 - 1}{C} \beta_{k-1} + h_k^T (y_d(k) - h_k \beta_{k-1}) \right). \end{aligned} \tag{18}$$

From this, we can derive the recursive formula for the output weights that incorporates a forgetting mechanism. Similarly, when the $(N_0 + i_2)$th dataset is fed into the model, we have:

$$\beta_{k+i_2} = \left( H_{k+i_2}^T \Theta_{k+i_2}^T \Theta_{k+i_2} H_{k+i_2} + \frac{E}{C} \right)^{-1} H_{k+i_2}^T \Theta_{k+i_2}^T \Theta_{k+i_2} Y_d(k+i_2). \tag{19}$$

In Eq. (19),

$$\Theta_{k+i_2} = \text{diag}\left\{ \sqrt{\prod_{s=0}^{i_2} \theta_{k+s}}, , \sqrt{\prod_{s=0}^{i_2} \theta_{k+s}}, \sqrt{\prod_{s=1}^{i_2} \theta_{k+s}}, \sqrt{\prod_{s=2}^{i_2} \theta_{k+s}}, \cdots, \sqrt{\prod_{s=i}^{i_2} \theta_{k+s}}, 1 \right\}.$$

The recursive formula for $\beta_{k+i_2}$ is:

$$\beta_{k+i_2} = \beta_{k+i_2-1} + \left( P_{k+i_2} + \frac{E}{C} \right)^{-1} \left( \frac{\theta_{k+i_2}^2 - 1}{C} \beta_{k+i_2-1} + h_{k+i_2}^T \left( y_d(k+i_2) - h_{k+i_2}\beta_{k+i_2-1} \right) \right) \tag{20}$$

In Eq. (20), $P_{k+i_2} = \theta_{k+i_2}^2 H_{k+i_2-1}^T H_{k+i_2-1} + h_{k+i_2}^T h_{k+i_2} = \theta_{k+i_2}^2 P_{k+i_2-1} + h_{k+i_2}^T h_{k+i_2}$.

### Dynamic adjustment strategy for model structure

Online adjustment of output layer parameters helps the model adapt to new data. However, as operational conditions change and data distributions shift, the neural network may struggle to handle new data characteristics. To address this, a dynamic structural adjustment strategy based on network pruning is proposed. This strategy optimizes the model structure and parameters, enhancing the adaptability of Stochastic Configuration Networks.

Assuming that a regularized stochastic configuration network with L hidden layer nodes has been constructed based on the training data set $N_0$, the output of the neural network is given by:

$$F_{L,0}^T(X_d(k-1)) = \sum_{j=1}^{L} \beta_{j,0} g_{j,0}(\omega_{j,0}^T X_d^T(k-1) + b_{j,0}). \tag{21}$$

In Eq. (21), $F_{L,0}^T(X_d(k-1))$ represents the output of a regularized stochastic configuration network without network structure adjustment, while $\beta_{j,0}, g_{j,0}, \omega_{j,0}, b_{j,0}, j$ represents the output weight of the hidden node $j$, the activation function of the hidden node $j$, the input weight of the hidden node $j$, and the bias of the hidden node $j$, respectively. When new data flows into the model and the accuracy remains unsatisfactory after updating the parameters of both the linear and nonlinear models, an adjustment of the structure of the nonlinear model is necessary. The adjustment criterion can be described as follows:

$$\left\| F_{L,0}^T(x_d(k)) - \Delta\bar{\rho}(k) \right\| > 3\sigma(k). \tag{22}$$

When the difference between the estimated values of the nonlinear model and the nonlinear labels exceeds three standard deviations, the model structure requires a dynamic adjustment. After pruning the $I$th hidden node, the model output is:

$$F_{L-1,1}^{'T}(x_d(k)) = \sum_{j=1}^{L} \beta_{j,0} g_{j,0}(\omega_{j,0}^T x_d^T(k) + b_{j,0}) - \beta_{I,1}' g_{I,1}'(\omega_{I,1}^{'T} x_d^T(k) + b_{I,1}'). \tag{23}$$

Thus, the change in network residual can be expressed as:

$$\Delta F_I = \left\| Y_d(k) - F_{L-1,1}^{'T}(x_d(k)) \right\|. \tag{24}$$

By comparing the impact of each hidden layer node on the change in model output residuals and sorting them by $\Delta F_{(1)} < \Delta F_{(2)} < \cdots < \Delta F_{(L)}$, we select and prune the $N_{prun}$ nodes with the least impact. The value of $N_{prun}$ satisfies $\Delta F_{(N_{prun})} / \left\| F_{L,0}^T(x_d(k)) \right\| < \sigma_p/L$ and $\Delta F_{(N_{prun}+1)} / \left\| F_{L,0}^T(x_d(k)) \right\| > \sigma_p/L$, where $\sigma_p$ is the pruning coefficient that determines the number of nodes to be pruned. After pruning, the output of the nonlinear model is represented as $F_{L-N_{prun},1}^{'T}(X_d(k))$, and the current network residual is as follows:

$$e_{L-N_{prun}}' = F_{L-N_{prun},1}^{'T}(X_d(k)) - Y_d(k) = \left[ e_{L-N_{prun},1}', e_{L-N_{prun},2}' \right]. \tag{25}$$

Incorporating new nodes based on the supervision mechanism. Subsequently, compute the output weights as follows:

$$\beta_{k,L-N_{prun}+1} = \left( H'_{k,L-N_{prun}+1} T \cdot H'_{k,L-N_{prun}+1} + \frac{E}{C} \right)^{-1} H'_{k,L-N_{prun}+1} T \Theta_k Y_d(k). \tag{26}$$

$H'_{k,L-N_{prun}+1} = [h'_{k,1}, h'_{k,2}, \ldots, h'_{k,L-N_{prun}+1}]$, $h_{k,L-N_{prun}+1}$ denotes the hidden layer output of the nonlinear model at the $(L - N_{prun} + 1)$th hidden node trained using dataset $N_0 + 1$. After incorporating the new nodes, the network output is:

$$F^T_{L-N_{prun}+1,1}(X_d(k)) = \sum_{j=1}^{L-N_{prun}+1} \beta_{j,1} g_{j,1}(\omega^T_{j,1} X^T_d(k) + b_{j,1}). \tag{27}$$

Next, determine if the network output error meets the predefined error criteria. If the criteria are satisfied, the model construction is complete; otherwise, new hidden layer nodes will be added based on a supervisory mechanism to minimize the output error until the termination condition is met.

## Adaptive intelligent detection method for slurry density based on collaborative computing

With the rapid advancement of Internet of Things (IoT) technology, we have entered an era of ubiquitous connectivity. Innovations such as cloud computing, big data, and artificial intelligence are transforming industrial applications through Internet platforms. In this context, edge-cloud collaboration has emerged as a crucial technology. Unlike traditional frameworks, edge computing enhances data processing by performing initial tasks near the data source (*e.g.*, equipment or sensors). Edge devices handle data acquisition and preliminary analysis, while edge control systems conduct initial data processing. This reduces the burden on central cloud servers, improving processing speed and efficiency. By addressing the limitations of traditional edge-cloud collaboration in real-time data processing, this approach enables efficient, real-time analysis and decision-making (*Zhou et al., 2021*). Edge-cloud collaboration has advanced industrial automation and intelligence, laying a strong foundation for Industry 4.0.

As illustrated in Fig. 1, the proposed online intelligent detection method for slurry density uses an edge-cloud collaborative framework to enhance real-time monitoring and intelligent analysis. Edge devices acquire and preprocess data, ensuring system stability and responsiveness. The edge control system processes data, runs online detection models, and allows operators to monitor key parameters such as slurry pump current, frequency, pressure, and density in real time. Operators can also input manual assay values *via* an interactive interface for model updates. The edge system's low latency and real-time capabilities meet the demands of industrial environments. Meanwhile, the cloud platform provides centralized computing power, managing databases and running slurry density detection software. It updates the initial model offline or online and deploys the updated model back to the edge for real-time detection. This architecture leverages the cloud's robust resources for iterative model optimization and centralized data management.

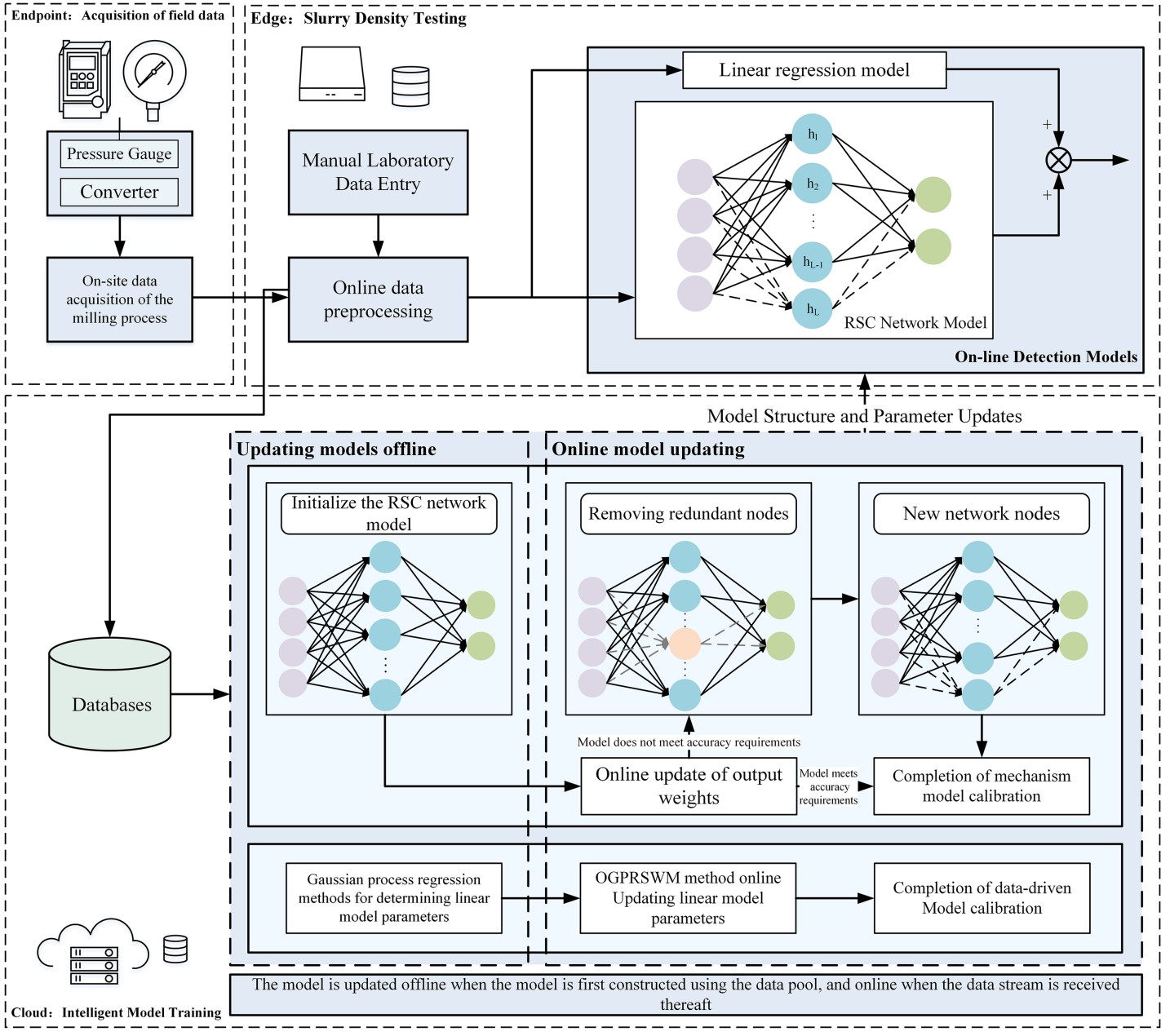

**Figure 1  Collaborative computing-based pulp density intelligent detection system structure diagram.**

## EXPERIMENTAL ANALYSIS

The process data in this study were collected from the grinding and classification stages of an actual mineral processing operation using industrial instruments. High-pressure sensors, low-pressure sensors, motor current, and motor voltage transmitted data *via* 4-20 mA signals to a Siemens S7-1500 PLC. The PLC used the Modbus-RTU protocol to communicate with edge servers, transferring real-time field data. These data captured

various operating conditions, such as changes in raw ore properties, equipment aging, and fluctuations in process parameters. Manual data were obtained through periodic on-site assays, covering slurry densities ranging from 1,000 to 1,500 kg/m$^3$. Variations were influenced by operational changes, such as the addition of ore or water. Measurement errors, caused by instrument limitations and environmental factors, were inevitable. To improve model performance, significant outliers were removed by cross-referencing with manual assay results. The cleaned dataset contained 800 samples, split in a 1:3 ratio for initial model training and streaming data. Min-Max Normalization was applied to remove dimensional unit interference and standardize the data for model training. Let $X = [P_H, P_L, f, I]$, and the data were processed using Eq. (28):

$$X' = \frac{X - \min(X)}{\max(X) - \min(X)}. \tag{28}$$

The dataset exhibited both sudden and gradual concept drift. Sudden drift resulted from abrupt changes, such as ore property variations, equipment failures, or emergency operational adjustments. Gradual drift arose from factors like equipment aging or long-term parameter fine-tuning. During the evaluation phase, the dataset was fed sequentially into the model as a data stream, maintaining the chronological order of collection. After processing each data point, the estimation error was calculated, and the model was updated. RMSE and MAE were computed cumulatively to compare different models' performance, demonstrating the proposed method's robustness under various drift scenarios.

The initial model was trained using two offline learning methods: GPR for the mechanistic model and RSC Network for the data-driven model. Once trained, the model is not further updated. The model estimates' results are shown in Fig. 2, with absolute errors in Fig. 3 and relative errors in Fig. 4. In the first 180 samples, conditions were relatively stable, and the model achieved high accuracy, with most absolute errors under 10 and relative errors below 1%. However, for samples 180–200, significant operational changes led to poor estimates, suggesting the model failed to capture new data distribution features. In the remaining dataset, the model's performance deteriorated further, highlighting the need for continuous learning to address frequent changes in operational conditions. This degradation reflects a concept drift phenomenon. To mitigate this, we propose an algorithm enabling online updates to adapt quickly to new distributions, ensuring high performance in industrial applications.

To demonstrate the effectiveness and superiority of the proposed intelligent detection method for concept drift data streams, we compared our method, OGPRSWM-FMRSC, with several other models. The linear model used is the online Gaussian process regression with sliding window mechanism (OGPRSWM), and the nonlinear model is a Regularized Stochastic Configuration Network (RRCN) with a forgetting mechanism. The alternative models evaluated include OGPR-FMRSC, which uses a standard Online Gaussian Process Regression (OGPR) without the sliding window, retaining historical data. The key

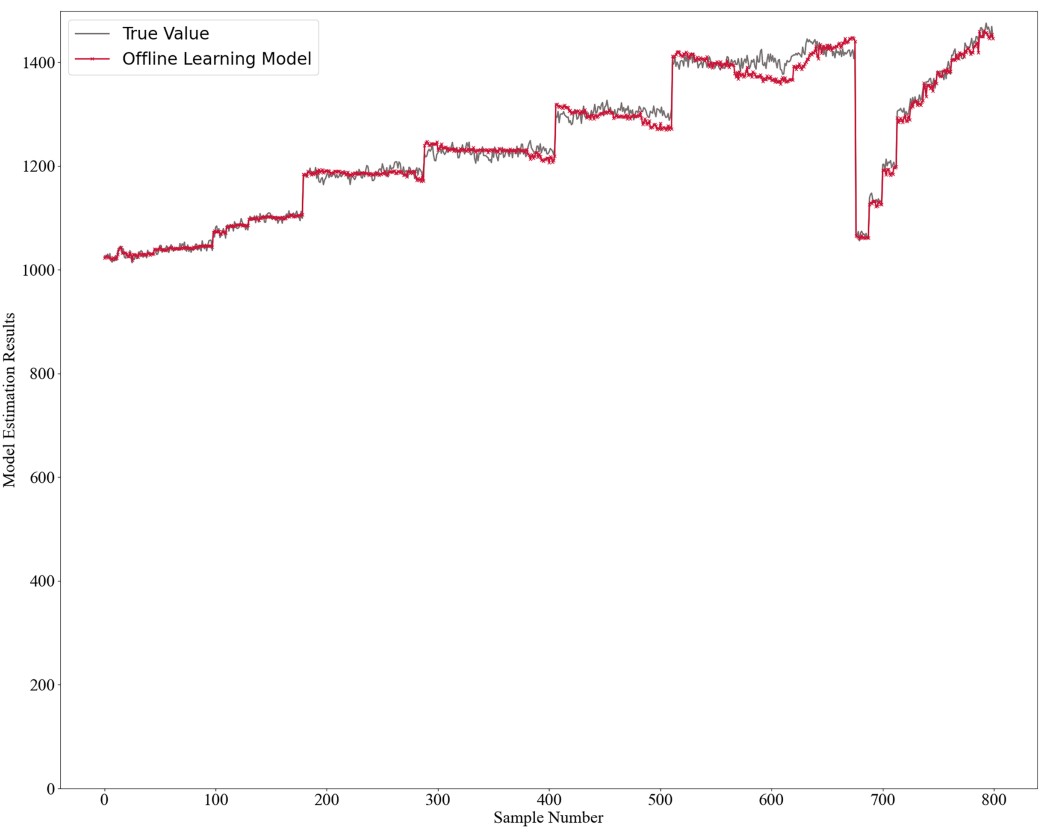

**Figure 2 Offline learning model estimation results.**

parameter update formula is shown in Eq. (29), and the nonlinear model is the same as our proposed algorithm.

$$
\begin{aligned}
A(k) &= \Sigma^{-1} + \sigma_n'^{-2} X_b' T(k) X_b'(k) \\
&= \Sigma^{-1} + \sigma_n'^{-2} \sigma_n^2 \big(A(k-1) - \Sigma^{-1}\big) + \sigma_n'^{-2} x_b' T(k) x_b'(k).
\end{aligned}
\tag{29}
$$

OGPRSWM-OSSC uses OGPRSWM for the linear model and an Online Sequential Stochastic Configuration Network (OSSC) for the nonlinear model. OGPRSWM-OSRSC utilizes OGPRSWM for the linear model and an Online Sequential Regularized Stochastic Configuration Network (OSRSC) for the nonlinear model. The output weights are updated online as follows:

$$
\beta_k = \beta_{k-1} + \left( P_k + \frac{E}{C} \right)^{-1} h_k^T (y_d(k) - h_k \beta_{k-1}).
\tag{30}
$$

In Eq. (30), $P_k = H_{k-1}^T H_{k-1} + h_k^T h_k = P_{k-1} + h_k^T h_k$; OGPRSWM-FWRSC incorporates OGPRSWM for the linear model and updates the output weights of the nonlinear model using the proposed online update method without dynamic structural adjustments. We evaluated the models using metrics such as $R^2$, minimum error frequency, $P_{\delta<1.0\%}$, MAE, RMSE, true positive rate (TPR), true negative rate (TNR), and mean relative error (MRE).

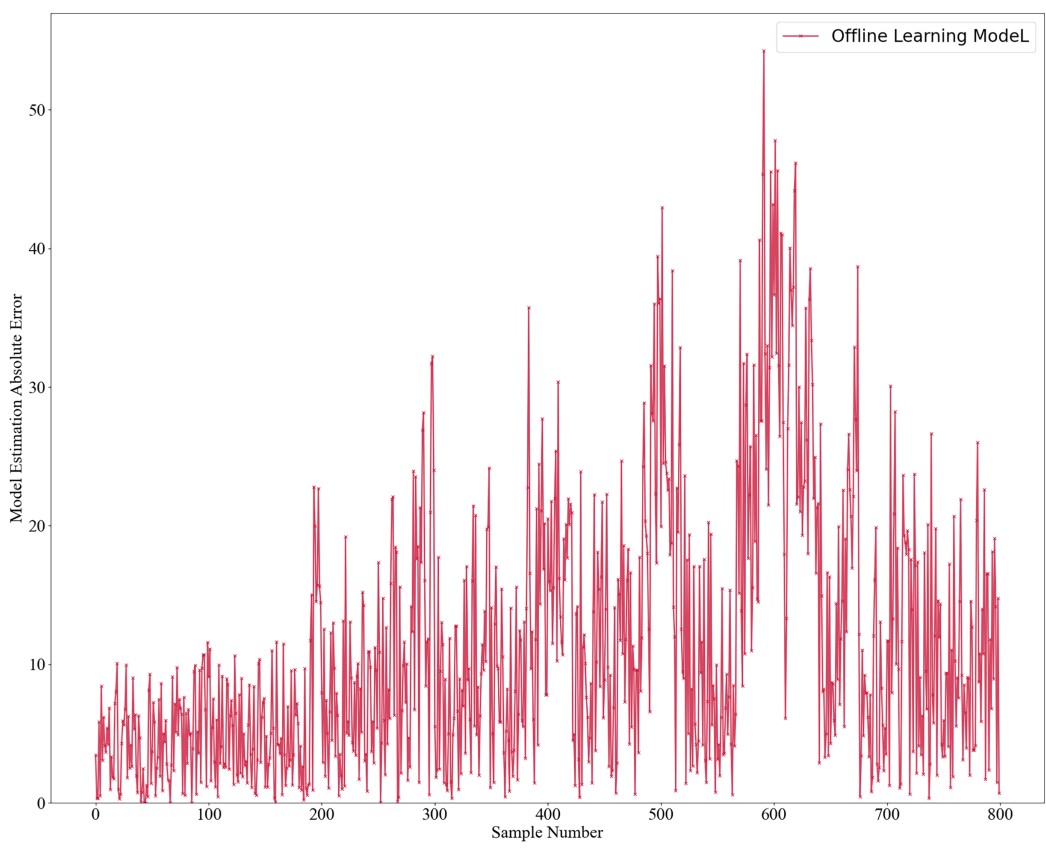

**Figure 3 Absolute error estimation of offline learning model.**

Figure 5 compares $R^2$, minimum error frequency, TPR, TNR, and A across different models, while Fig. 6 compares RMSE and MAE. Overall, OGPRSWM-FMRSC outperformed the other models in all metrics. Specifically, the sliding window mechanism in OGPRSWM-FMRSC proved effective in handling concept drift, as evidenced by its superior performance compared to OGPR-FMRSC. The comparative performance of OGPRSWM-FMRSC, OGPRSWM-FWRSC, OGPRSWM-OSRSC, and OGPRSWM-OSSC sequentially declined, highlighting the importance of dynamic structure adjustment and the combination of the forgetting mechanism with regularized least squares in enhancing model performance.

Table 1 presents the performance evaluation metrics of the five models for slurry density detection. The initial condition of the test dataset is labeled as Condition 1, with subsequent significant density changes due to sample addition labeled as Conditions 2, 3, and 4. Condition 5 begins around sample number 680, reflecting multiple sample additions over a short period. Table 2 shows the MRE of each model under different conditions. OGPRSWM-FMRSC demonstrated superior performance, with the lowest MAE and RMSE of 6.11 and 7.56, respectively. Its $R^2$ value reached 99.40%, indicating a high fit between the model's estimates and actual data. The OGPRSWM-FMRSC model

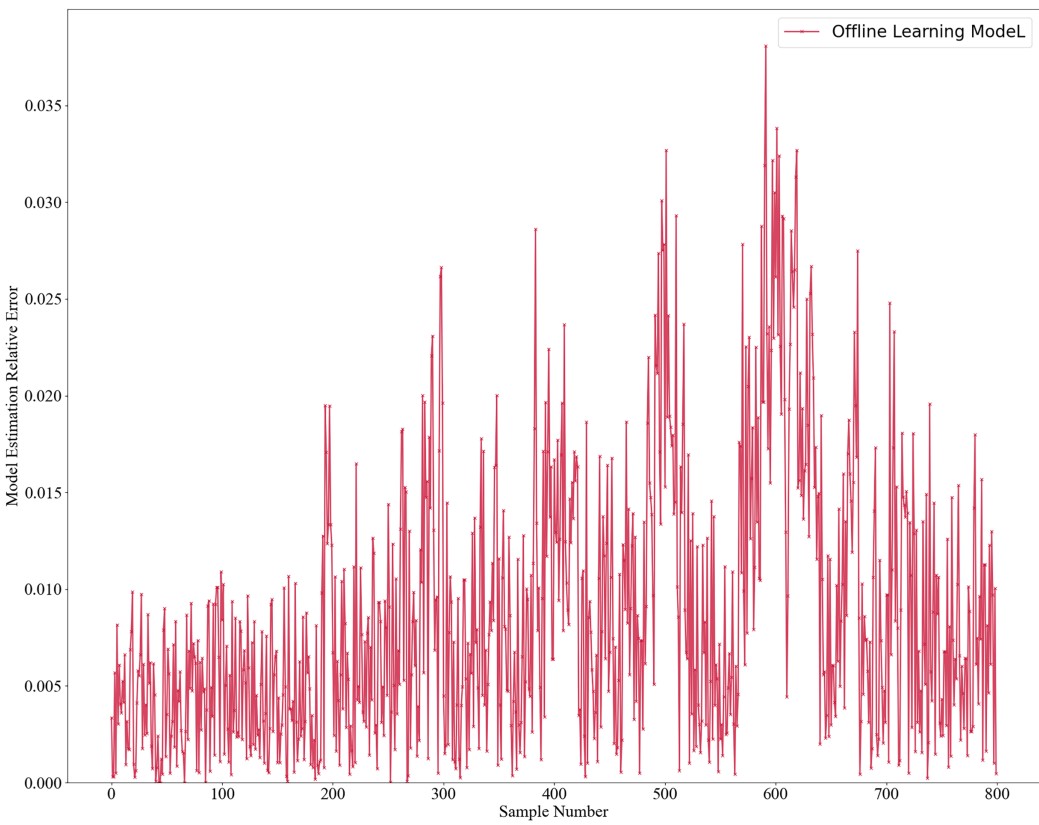

**Figure 4** **Relative error estimation of offline learning model.**

also had 91% of samples with a relative error below 1.0%, and its TPR and TNR were 75.88% and 76.39%, respectively, outperforming the other models.

Figure 7 presents the probability density function of estimation errors for the five models. The error distribution for OGPRSWM-FMRSC is centered around zero and exhibits a unimodal peak consistent with Gaussian distribution characteristics, suggesting that the error sequence approximates randomness. Figure 8 presents the autocorrelation function of OGPRSWM-FMRSC's error, indicating that it approaches white noise levels, with errors primarily attributed to random factors rather than poor model generalization. This suggests that OGPRSWM-FMRSC has superior estimation and generalization capabilities, making it more suitable for dynamic industrial environments and potentially more stable under specific conditions compared to the other models.

## INDUSTRIAL APPLICATION ANALYSIS

In industrial applications, the Siemens S7-1500 PLC interfaces with edge devices *via* the RS485 bus and Modbus-RTU protocol to collect and transmit real-time field data. Edge devices utilize TIAV16 and Modscan32 software to simulate Modbus communication,

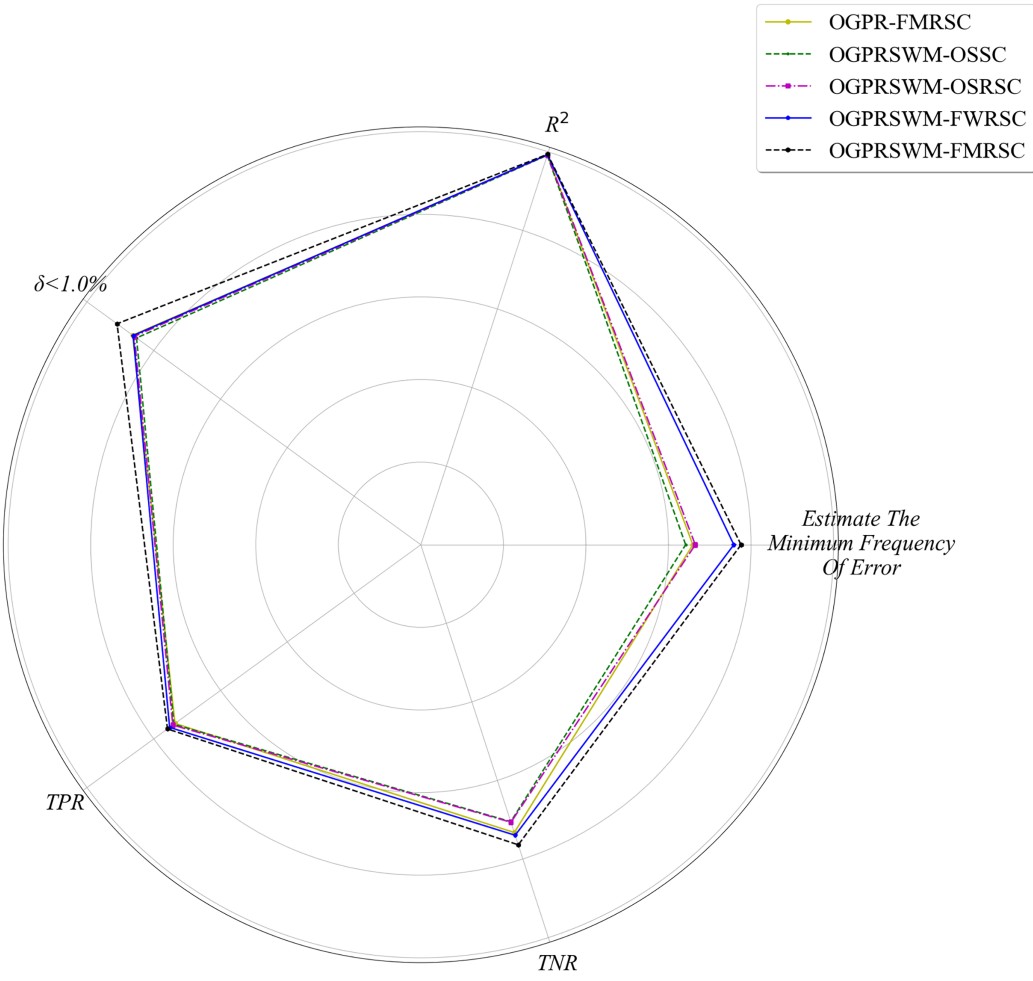

**Figure 5 Comparison of different models in terms of $R^2$, minimum estimation error frequency, TPR, TNR and $P_{\delta<1.0\%}$.**

enabling remote monitoring and control of field equipment. The edge data is transmitted to the cloud for analysis and storage using a proprietary cloud protocol, as illustrated in Fig. 9. To enable online slurry density detection and provide a intuitive interface, a software application based on Vue, Spring Boot, and Flask frameworks was developed. This software supports data visualization, storage, and query functions. It has been deployed for over 5 months at a beneficiation plant in Shenyang. The interface design and human-machine interaction prioritize efficiency and ease of use, optimizing operational procedures, reducing operational difficulty, and significantly reducing the frequency of operator errors, thereby enhancing operational safety and production efficiency. As shown in Fig. 10, the real-time slurry density detection module displays the slurry density trend calculated by the intelligent detection model alongside scatter points representing manually obtained density values. By hovering the mouse over any data point reveals the specific slurry density value at that point. Figure 11 illustrates a table from the software interface, showing the most recent nine sets of comparison values obtained through random sampling and testing post-system deployment. In these nine sets, the relative error

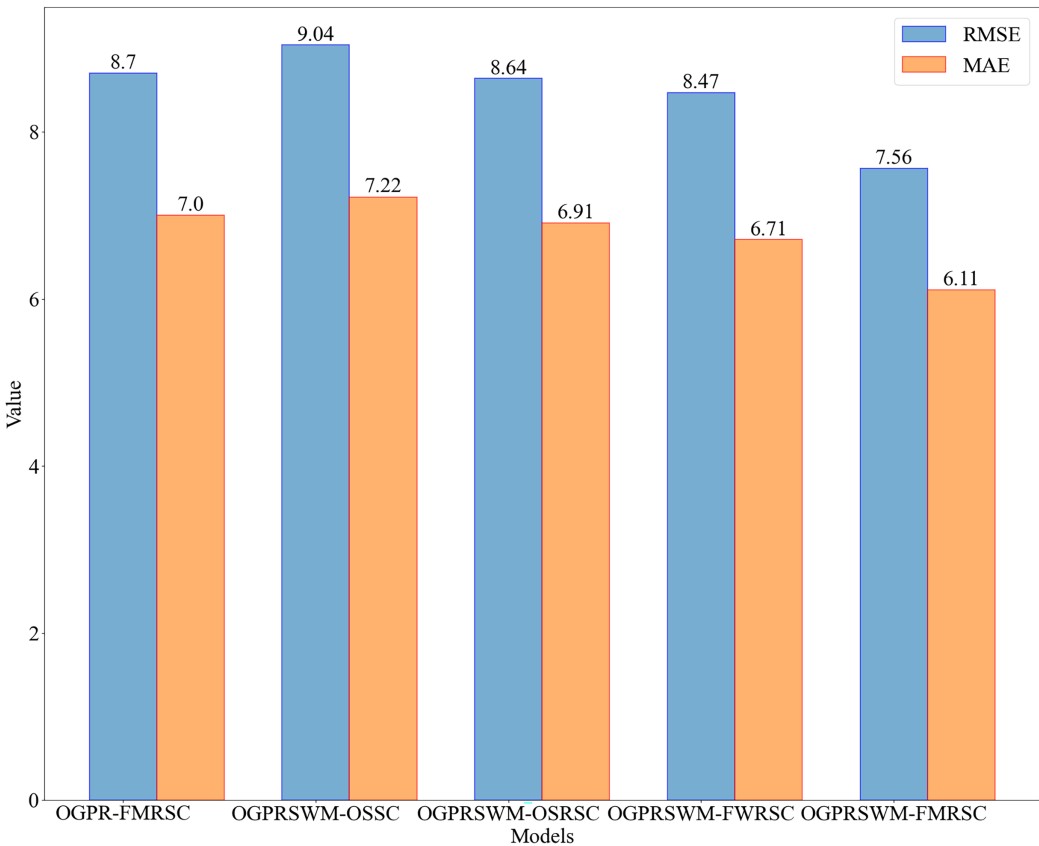

**Figure 6  Comparison of RMSE and MAE among different models.**

**Table 1  Model evaluation results.**

| Model | Estimate the minimum frequency of error | MAE | RMSE | TPR% | TNR% | $R^2$% | $P_{\delta < 1.0\%}$ |
|---|---|---|---|---|---|---|---|
| OGPRSWM-FMRSC | 22.17% | 6.11 | 7.56 | 75.88% | 76.39% | 99.40% | 91.00% |
| OGPR-FMRSC | 18.83% | 7.00 | 8.70 | 73.63% | 73.26% | 99.20% | 86.33% |
| OGPRSWM-OSSC | 18.33% | 7.22 | 9.04 | 73.95% | 70.49% | 99.14% | 85.16% |
| OGPRSWM-OSRSC | 19.00% | 6.91 | 8.64 | 74.28% | 70.64% | 99.22% | 85.83% |
| OGPRSWM-FWRSC | 21.67% | 6.55 | 8.27 | 75.24% | 73.96% | 99.28% | 86.16% |

between the estimated slurry density and the actual test results did not exceed 1%. Figure 12 illustrates a bar chart distribution of the slurry density estimation errors over the 5 months of operation, and Table 3 details the corresponding error analysis data. With an acceptable relative error threshold of less than 2%, all months showed a qualification rate above 95%, indicating that the proposed adaptive intelligent detection system based on collaborative computing performed effectively in industrial settings, significantly enhancing production efficiency.

**Table 2 Model evaluation for MRE under different operating conditions.**

| Model | Operating conditions 1 | Operating conditions 2 | Operating conditions 3 | Operating conditions 4 | Operating conditions 5 |
|---|---|---|---|---|---|
| Offline Model | 0.7983% | 1.3347% | 1.6825% | 2.0699% | 2.4299% |
| OGPRSWM-FMRSC | 0.5269% | 0.5072% | 0.4656% | 0.4383% | 0.4507% |
| OGPR-FMRSC | 0.6253% | 0.4927% | 0.5720% | 0.4864% | 0.5691% |
| OGPRSWM-OSSC | 0.6276% | 0.5428% | 0.5614% | 0.5315% | 0.5499% |
| OGPRSWM-OSRSC | 0.5975% | 0.5076% | 0.5694% | 0.5087% | 0.5133% |
| OGPRSWM-FWRSC | 0.5439% | 0.6166% | 0.5499% | 0.4394% | 0.4954% |

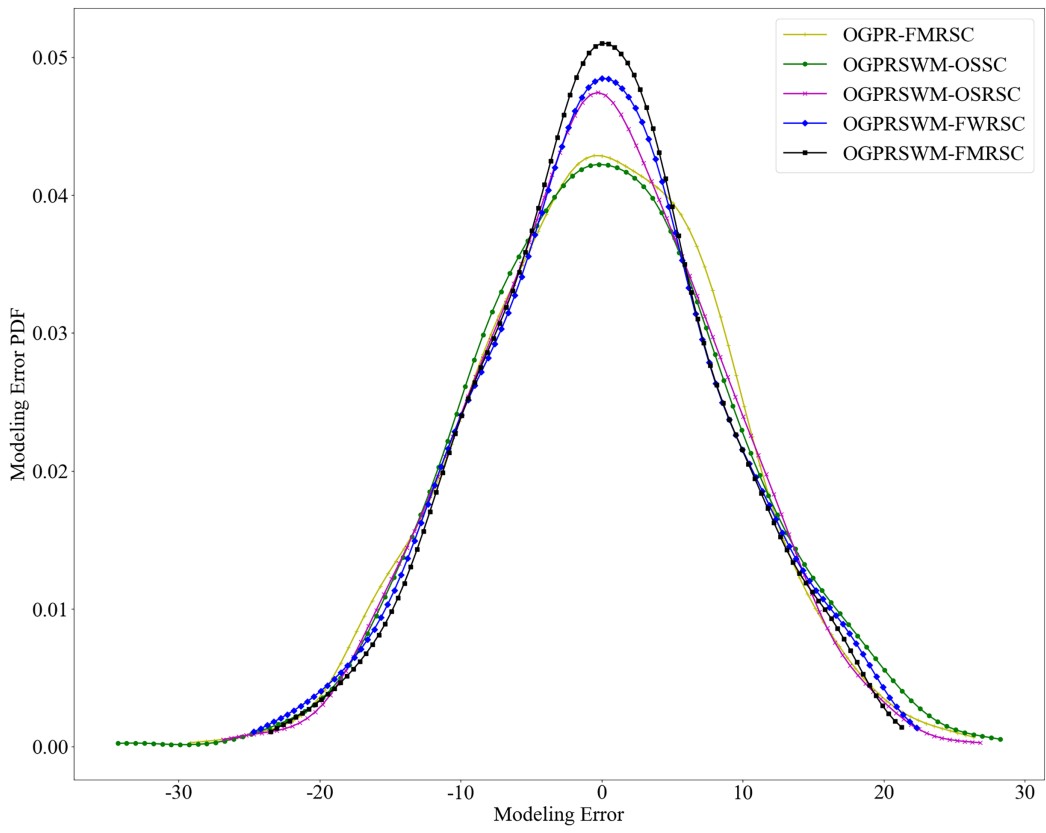

**Figure 7 Comparison of estimation error PDFs for different models.**

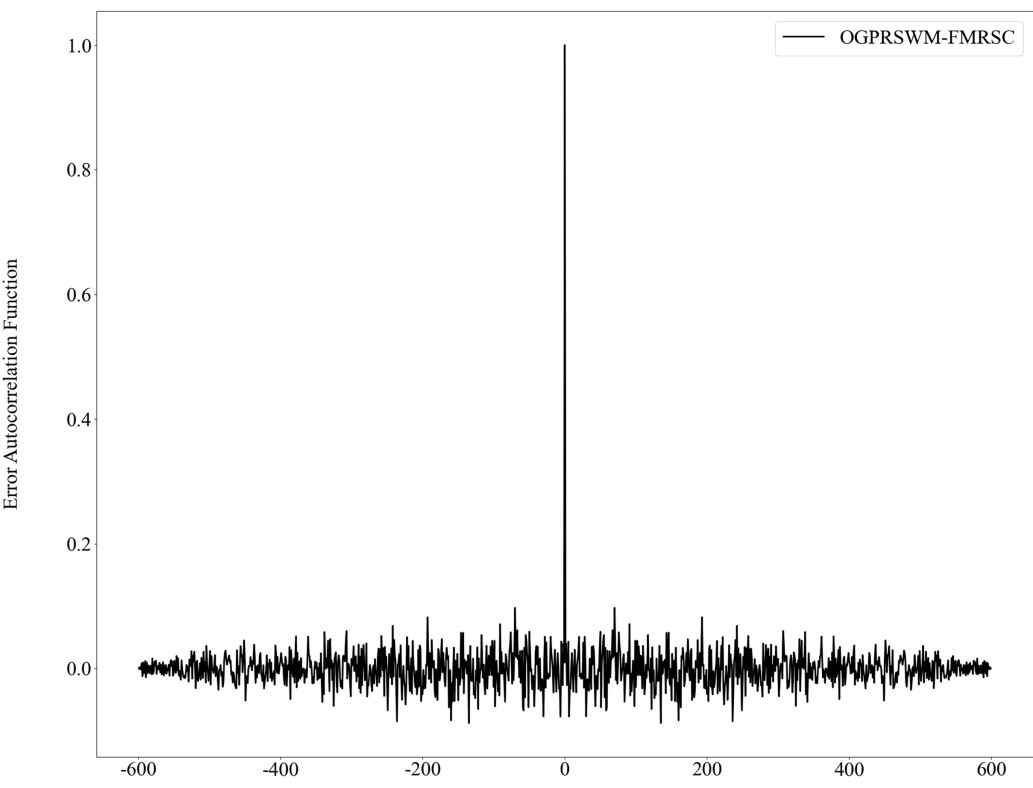

**Figure 8** Self-correlation function of estimation error for the OGPRSWM-FMRSC model.

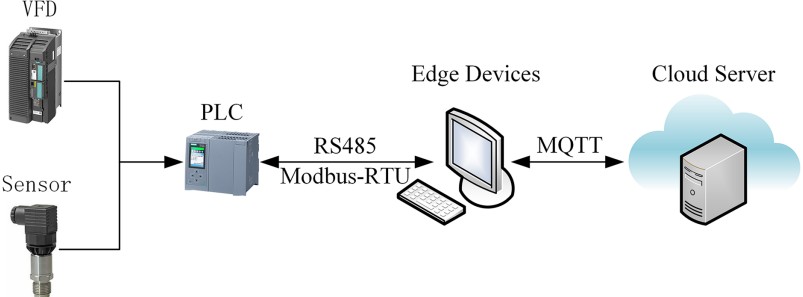

**Figure 9** Hardware platform framework.

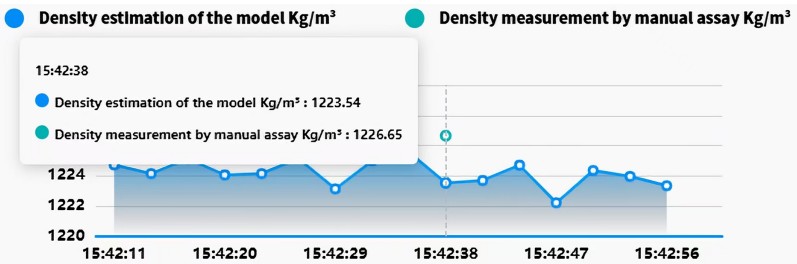

**Figure 10** Pulp density real-time monitoring module demonstration.

| 📅 | Start time | **to** | End time | | Please select a range of parameters and values ⌄ | | 🔍 Inquire | | Generate reports |

| Acquisition time ⇕ | Whether it is abnormal | High pressure value Pa ⇕ | Low pressure value Pa ⇕ | Inverter current A ⇕ | Inverter frequency Hz ⇕ | The model estimates the density Kg/m³ ⇕ | Density of manual assays Kg/m³ ⇕ |
|---|---|---|---|---|---|---|---|
| 2023/12/04 15:42:38 | No | 37231.21 | 25141.51 | 3.25 | 41.42 | 1223.54 | 1226.65 |
| 2023/12/04 14:26:25 | No | 51922.41 | 39079.48 | 3.6 | 48.82 | 1305.21 | 1311.78 |
| 2023/12/04 13:04:27 | No | 50654.66 | 37875.03 | 3.64 | 55.92 | 1296.42 | 1300.52 |
| 2023/12/04 12:00:10 | No | 52973.14 | 40801.15 | 3.58 | 50 | 1304.68 | 1301.55 |
| 2023/12/04 11:03:35 | No | 29603.94 | 17590.19 | 3.1 | 35.94 | 1225.89 | 1229.12 |
| 2023/12/04 10:04:39 | No | 30717.96 | 19115.65 | 3.12 | 37.73 | 1183.12 | 1183.90 |
| 2023/12/04 09:11:56 | No | 28325.34 | 16729.36 | 3.09 | 35.9 | 1195.99 | 1189.26 |
| 2023/12/04 08:31:36 | No | 28084.81 | 16482.5 | 3.1 | 35.91 | 1182.12 | 1183.90 |
| 2023/12/04 08:10:24 | No | 28097.47 | 16488.83 | 3.11 | 35.88 | 1178.18 | 1184.55 |

**Figure 11 Sampling inspection result table.**

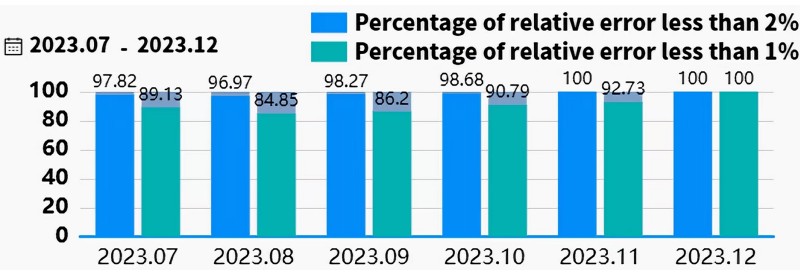

**Figure 12 Bar chart of the pulp density estimation error after more than 5 months of operation.**

**Table 3 Analysis of the pulp density estimation error after more than 5 months of operation.**

| Month | Monthly Laboratory Tests | Average Density (Manual Laboratory Test) | Average Absolute Error | Maximum Absolute Error | $P_{\delta<1.0\%}$ | $P_{\delta<2.0\%}$ (Satisfactory Rate) |
|---|---|---|---|---|---|---|
| 2023.07 | 92 | 1,311.52 | 7.32 | 30.04 | 89.13% | 97.82% |
| 2023.08 | 33 | 1,295.63 | 6.98 | 27.43 | 84.85% | 96.97% |
| 2023.09 | 58 | 1,310.46 | 7.54 | 34.76 | 86.20% | 98.27% |
| 2023.10 | 76 | 1,255.34 | 6.91 | 25.78 | 90.79% | 98.68% |
| 2023.11 | 55 | 1,287.64 | 7.11 | 19.94 | 92.73% | 100.00% |
| 2023.12 | 9 | 1,234.58 | 3.98 | 6.73 | 100.00% | 100.00% |

## CONCLUSION

This study addresses concept drift in slurry density detection models within industrial environments and proposes an intelligent detection algorithm for concept drift data streams. Operational changes over time often lead to a gradual decline in model performance. To address this, a sliding window mechanism is incorporated into the linear model, with recursive formulas derived for real-time parameter updates. This approach minimizes the impact of outdated data on current model accuracy. For nonlinear models, a forgetting mechanism is introduced, with recursive formulas developed for online updating of output weights, reducing the influence of historical data on new detections. Additionally, network pruning and stochastic configuration methods are used to optimize the model structure, enhancing its adaptability to new data distributions. Weighted least squares and regularization methods are integrated during the stochastic configuration process to evaluate output weights, improving the model's generalization capabilities. Experimental results show that the proposed method achieves superior accuracy and stability when handling concept drift, significantly improving the reliability of slurry density detection. This research has both academic significance and industrial value. The real-time update algorithm enhances slurry density detection precision and stability, providing an efficient monitoring tool for production processes. Accurate slurry density detection is vital for optimizing process parameters, improving coal preparation accuracy, and minimizing resource waste. By quickly responding to operational changes, the proposed method prevents fluctuations in concentrate quality caused by detection errors, improving production controllability. Its low computational cost makes it suitable for real-time industrial applications, while also reducing resource waste and the need for frequent manual adjustments or shutdown maintenance in high-frequency production scenarios. Beyond slurry density detection, the proposed model framework is versatile and can be applied to other industrial domains. For example, it can be used for sensor data monitoring in equipment fault prediction and dynamic load regulation in energy management. By addressing concept drift effectively, this method adapts to various complex industrial scenarios, providing robust support for intelligent manufacturing in the Industry 4.0 era.

### Funding

This work was financially supported by the National Natural Science Foundation of China (52304309) and the Fundamental Research Funds for the Central Universities of China (2023QN1086). The funders had no role in study design, data collection and analysis, decision to publish, or preparation of the manuscript.

### Grant Disclosures

The following grant information was disclosed by the authors:
National Natural Science Foundation of China: 52304309.
Fundamental Research Funds for the Central Universities of China: 2023QN1086.

### Competing Interests

The authors declare that they have no competing interests.

### Author Contributions

- Lanhao Wang conceived and designed the experiments, performed the experiments, analyzed the data, performed the computation work, prepared figures and/or tables, authored or reviewed drafts of the article, and approved the final draft.
- Hao Wang conceived and designed the experiments, performed the experiments, analyzed the data, performed the computation work, prepared figures and/or tables, authored or reviewed drafts of the article, and approved the final draft.
- Taojie Wei conceived and designed the experiments, performed the experiments, analyzed the data, performed the computation work, prepared figures and/or tables, authored or reviewed drafts of the article, and approved the final draft.
- Wei Dai conceived and designed the experiments, performed the experiments, analyzed the data, performed the computation work, prepared figures and/or tables, authored or reviewed drafts of the article, and approved the final draft.
- Hongyan Wang conceived and designed the experiments, performed the experiments, analyzed the data, performed the computation work, prepared figures and/or tables, authored or reviewed drafts of the article, and approved the final draft.

### Data Availability

The raw data and code are available in the Supplemental Files.

### Supplemental Information

Supplemental information for this article can be found online at http://dx.doi.org/10.7717/peerj-cs.2683#supplemental-information.

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
