# Peer review of "An online intelligent detection method for slurry density in concept drift data streams based on collaborative computing"

_PeerJ Computer Science, doi:10.7717/peerj-cs.2683_

## Round 0.1 · original submission · Minor Revisions

Dear authors,

Some concerns are still remaining that need to be addressed.

You are advised to critically respond to the comments point by point when preparing a new version of the manuscript and while preparing for the rebuttal letter. R2 has provided additional comments in their PDF

Kind regards,
PCoelho

Reviewer 1 ·

Basic reporting

Formula Display Errors: Some of the formulas in the text exhibit display issues, particularly symbol overlap and formatting inconsistencies, which compromise their clarity and accuracy, causing inconvenience for readers. Please thoroughly review and correct these errors to ensure that all formulas are presented correctly and accurately, thereby maintaining the scholarly rigor of the paper.

Experimental design

I noticed an excessive use of formulas throughout the article, which may pose difficulties for readers in comprehending the paper's message. While formulas are essential in scientific papers to precisely express mathematical relationships and theoretical models, an overwhelming number of them can confuse non-specialist readers and potentially detract from the grasp of the paper's core ideas. Therefore, I recommend maintaining only the necessary formulas while appropriately reducing their quantity.

Validity of the findings

no comment

Additional comments

As above

Reviewer 2 ·

Basic reporting

The work topic is sure of interest, and the amount of work done, and information provided define the work contributions. Also, the structure of the article is well done. Authors explained the contributions made in the paper very well, as well as the structure of other sections.

Experimental design

This submission considered the issue of concept drift in slurry density detection models within industrial environments and proposed an intelligent detection algorithm specifically designed for slurry density in concept drift data streams.

Validity of the findings

The work is clear, and the framework is straightforward. However, I have a few concerns after reviewing the submission, particularly regarding the validity of the findings. Please, check my comments, attached.

Annotated reviews are not available for download in order to protect the identity of reviewers who chose to remain anonymous.

---

## Round 0.2 · Minor Revisions

Dear authors,

Thanks a lot for your efforts to improve the manuscript.
Nevertheless, some concerns are still remaining that need to be addressed.
Like before, you are advised to critically respond to the remaining comments point by point when preparing a new version of the manuscript and while preparing for the rebuttal letter.

Kind regards,
PCoelho

Reviewer 1 ·

Basic reporting

1.Language and Grammar:
While the overall language is clear and professional, some sentences can be more fluent.
2.Literature References:
The references are comprehensive and relevant, but the inclusion of some recent studies would enhance the timeliness of the paper.
3.Structure:
The paper's structure conforms to PeerJ standards and is logically organized, making it easy to follow. However, more technical details in the methods section would help readers better understand the research process.

Experimental design

1.The research question is well-defined, relevant, and meaningful. The paper clearly states how the research addresses an identified knowledge gap.
2.The methods are described in sufficient detail to allow replication by other researchers. More details about data preprocessing, especially feature selection and data cleaning, would be beneficial.
3.The investigation was conducted to a high technical and ethical standard, with rigorous experimental design.

Validity of the findings

1.All underlying data are robust, statistically sound, and well-controlled.
2.The conclusions are well-stated, linked to the original research question, and limited to supporting results. A deeper discussion of the implications of the findings and their impact on industrial applications would be valuable.

Additional comments

1.Please ensure that all figures and images have not been inappropriately manipulated. Additionally, the metadata identifiers in the supplementary files should be more descriptive to be useful to future readers.
2.The authors should provide a more detailed explanation of how the proposed method handles different types of concept drift (sudden, gradual, incremental, and recurrent). This would help readers better understand the robustness and adaptability of the model in various industrial scenarios.

Reviewer 2 ·

Basic reporting

Clear

Experimental design

Clear

Validity of the findings

Clear

---

## Round 0.3 · Minor Revisions

Dear authors,

Thanks a lot for your efforts to improve the manuscript.

Nevertheless, some concerns are still remaining that need to be addressed.
Like before, you are advised to critically respond to the remaining comments point by point when preparing a new version of the manuscript and while preparing for the rebuttal letter.

Kind regards,
PCoelho

Reviewer 1 ·

Basic reporting

1. The paper has a complete structure and is detailed and substantial in content.
2. The overall language is quite professional, but some sentences could be simplified.
3. Some of the images in the paper are blurry; it is recommended to replace them with higher-definition ones, such as Figure 1.

Experimental design

1. The description of the data set production process used in the experiment is brief; it is recommended to provide a more detailed account of the production process.

Validity of the findings

1. The ideas presented in the paper are innovative and quite meaningful.
2. The research conclusions clearly respond to the original research question and are stated solely based on supportive results. Further discussion on the implications of these findings and their impact on industrial applications would enhance the depth and value of the discourse.

---

## Round 0.4 · accepted · Accept

Dear authors, we are pleased to verify that you meet the reviewer's valuable feedback to improve your research.

Thank you for considering PeerJ Computer Science and submitting your work.

Kind regards
PCoelho

Reviewer 1 ·

Basic reporting

1. The abstract can be further streamlined.
2. Most of the cited references are relatively dated. It is recommended to further explore the current research in this field.
3. The following articles are relevant to this field, and it is recommended that you consider citing them.
(1) Transfer Learning for Regression through Adaptive Gaussian Process
(2) The influence of slurry density on in situ density

Experimental design

No comment

Validity of the findings

No comment

Additional comments

No comment